# Dengue Surveillance System in Brazil: A Qualitative Study in the Federal District

**DOI:** 10.3390/ijerph17062062

**Published:** 2020-03-20

**Authors:** Marco Angelo, Walter Massa Ramalho, Helen Gurgel, Nayara Belle, Eva Pilot

**Affiliations:** 1Department of Health, Ethics & Society, Maastricht University, 6200 MD Maastricht, The Netherlands; nayarabelle@gmail.com (N.B.); eva.pilot@maastrichtuniversity.nl (E.P.); 2Nucleo de Medicina Tropical, Universidade de Brasilia, 70297-400 Brasilia, Brazil; walter.ramalho@gmail.com; 3Laboratorio de Geografia, Ambiente e Saúde da Universidade de Brasilia, 70904-970 Brasilia, Brazil; helengurgel@unb.br; 4Centre of Studies in Geography and Spatial Planning (CEGOT), University of Coimbra, 3004-530 Coimbra, Portugal

**Keywords:** surveillance, dengue, public health, underreporting, tropical diseases, qualitative research, infectious diseases, health information, urban health, health geography

## Abstract

Dengue’s increasing trends raise concerns over global health and pose a challenge to the Brazilian health system, highlighting the necessity of a strong surveillance system to reduce morbidity, mortality, and the economic burden of this disease. Although the Brazilian surveillance system reports more dengue cases than any other country, recent studies suggest that non-reported cases are the majority. The aim of the study is to explore the strengths and weaknesses of the Brazilian surveillance system, particularly looking at the functioning of data collection and reporting. This was done through qualitative semi-structured interviews with 17 experts in dengue surveillance, supported by quantitative data from the official notification system. To select the interviewees, purposive and theoretical sampling were used. Data were analyzed through thematic analysis. The research highlighted that a lack of human and technological resources in healthcare units and surveillance departments slows down the notification process and data analysis. Due to a lack of integration in the private sector, the surveillance system fails to detect the socioeconomic profile of the patients. Investments in public healthcare, human and technological resources for surveillance and better integration in the private healthcare system, and vector surveillance may improve dengue surveillance.

## 1. Introduction

### 1.1. Dengue, Dengue Surveillance, and Relevance in Global Public Health

Dengue is a viral mosquito-borne disease that can lead to severe and sometimes fatal consequences caused by Dengue Virus (DENV), of which five serotypes are known [1]. Dengue incidence has been steadily growing for the past 15 years, and dengue was labeled as the most important re-emerging mosquito-borne disease worldwide [2]. Due to its rapidly increasing incidence, dengue represents a major threat for the Brazilian health system [3].

With climate change, urbanization, and deforestation occurring globally and modifying the patterns of tropical diseases, dengue’s increasing incidence raises concerns for global health [4,5]. Dengue surveillance is crucial to the strategy of dengue prevention and control and the strengthening of surveillance is relevant for a global response to emerging infectious diseases [6]. Effective surveillance provides guidance for risk assessment and epidemic response, allowing the monitoring of disease trends and program evaluation [7]. To understand the spatial distribution of diseases and create the means to understand emerging health problems, disease surveillance is a relevant subject for health geography [8] and must be fully understood.

Existing literature shows how dengue cases are underreported globally, pointing out the necessity of improving data collection for surveillance [2]. Data collection for dengue is particularly complex due to the presence of five distinct serotypes and the multiplicity of data to collect, with World Health Organization (WHO) guidelines highlighting the importance of periodic data quality evaluation [6].

Epidemiological studies show how dengue appears predominantly in urban settings [9]. This is particularly relevant in the Brazilian setting, where the urban population forms an outstanding 86% of the total population, with urbanization still showing a growing trend [10]. In Brazil, dengue incidence is greater in cities with more than 500,000 inhabitants, highlighting the necessity for an adequate and effective urban surveillance system [11].

The most used form of surveillance is epidemiological data collection. This can be complemented by entomological, socioeconomic, and environmental data. Epidemiological data may be collected through active or passive collection processes. Passive surveillance relies on healthcare facilities to report the occurrence of diseases [12]. Despite passive surveillance being cost-effective and constituting the pillar of dengue surveillance, it is often associated with underreporting, especially of non-hospitalized cases [13]. The implementation of enhanced surveillance is thus recommended, consisting of the epidemiological analysis of reported data, syndromic surveillance, laboratory-based reporting, and active surveillance strategies [14].

The minimum set of indicators recommended from the WHO for dengue surveillance comprises: the number of suspected dengue cases, the number of severe dengue cases, the number of deaths from dengue, and the number of cases confirmed by the laboratory and serotypes in circulation [6]. For a timely response to epidemics, it is important to work with suspected cases, as case confirmation might be time consuming.

Laboratory confirmation is important for data collection, especially in areas where other arboviral infections co-exist [15]. A cost-effective, rapid diagnostic test for dengue is available, the NS1. The rapid test can be complemented by viral isolation or, more commonly, by PCR, which is a molecular technique used to detect the presence of the virus RNA. The PCR test can diagnose dengue up to 7 days after the onset and can be used to determine the serotype [16,17,18]. Through laboratory confirmation, surveillance can exclude the co-circulation of other arboviroses and detect circulating serotypes, as recommended in the International Health Regulations. The detection of new serotypes in one area represents an epidemic potential and is associated with an increase in dengue-related deaths [6,19].

Passive epidemiological surveillance can be complemented by active surveillance. A variety of active surveillance strategies exists, such as syndromic surveillance, sentinel surveillance, and active case finding. Syndromic surveillance is the monitoring of indicators for the early detection of outbreaks, such as school/work absenteeism, medication sales, and internet-based health inquiries. Sentinel surveillance is conducted through selected reporting units with a high probability of seeing dengue cases, such as hospitals and airports, which are entrusted to identify and notify the cases. Active case finding is the active research of cases in hospitals and healthcare units [20].

Complementary to epidemiological surveillance, environmental surveillance is recommended for vector control [6]. This requires considering numerous multisectoral aspects, as many environmental and socioeconomic factors have been linked to the breeding of A. Aegypti. These can be related to the climate and man-made urban environments, such as the presence of stagnant water in houses and abandoned buildings, and inadequate water, sanitation systems, and waste management [21,22,23].

### 1.2. Dengue Surveillance in Brazil

The national health surveillance system in Brazil is part of the national unified health system, which is coordinated by the Ministry of Health. As Brazil is a federal country, the surveillance system is organized on a municipal, state and federal level. The core capacities of the national health system fulfill most of the requisites listed in the International Health Regulations with respect to structure, surveillance, and response procedures, particularly on the national and state level. However, it has been noted how the current level of investment from the government in surveillance does not appear sufficient, and the results of the Brazilian surveillance system on dengue control are poor when compared to other infectious diseases [24,25].

Brazil has one of the most comprehensive dengue surveillance systems, with a multiplicity of data across 5570 municipalities [26]. The collected data include epidemiological data, weather patterns, entomological data, and socioeconomic and environmental data [27]. Epidemiological surveillance is mainly based on passive surveillance through the notification system for infectious diseases [28], which is the SINAN (Sistema de Informação de Agravos de Notificação). Dengue is on the national list of compulsorily notifiable diseases and can be notified online in the SINAN portal [29].

Although Brazil reports more dengue cases than any other country due to its efficient reporting system, research suggests that non-reported cases are the majority. A recent study in the Brazilian city of Salvador estimated that, for each 12 cases of dengue, only one was reported to the SINAN [30]. Another study revealed underreporting when comparing the hospitalized dengue cases registered in the records of the national health system to the cases collected through the SINAN, and also suggested underreporting from the private healthcare sector [31].

Private sector underreporting is particularly relevant as Brazilian healthcare relies on a growing number of private-public partnerships, and private healthcare is growing [32]. The lack of resources impedes the access to public healthcare [33,34], and the perceived quality of service is higher in the private sector. This influences Brazilian’s health seeking behavior, even in poor urban settings [35].

To investigate the functioning of dengue surveillance in Brazil, the surveillance system of the Federal District (Distrito Federal (DF)) will be evaluated as a case study. The DF is the federal unit where the Brazilian capital (Brasilia) is located. Contrary to the other Brazilian states, the DF has no municipalities, as the capital city constitutes the entire territory of the DF. Brasilia is the third largest city in Brazil and its territory is endemic for dengue, which disproportionally affects urban areas. Being the location of the national Ministry of Health, its setting allows for interviewing experts on both a federal and local level.

In the DF, dengue typically has a seasonal incidence, with an epidemic period during the warm season (October–May) and an inter-epidemic period in the cold season (June–September) [3]. In the last decade, dengue incidence in the DF has been steadily growing [36,37]. In the year of writing, the greatest dengue epidemic ever recorded in the DF occurred, with 47,745 notified cases between January and September 2019 and an incidence coefficient of 1340.50 cases per 100,000 inhabitants. The epidemic strained the healthcare facilities, requiring the implementation of public health emergency measures to contain the occurrence of dengue-related deaths [38].

### 1.3. Aim of the Study

Using the DF as a case study, the aim of the research is to analyze how the Brazilian Dengue surveillance system operates, focusing on the functioning of data collection and reporting, and on the integration of epidemiological and environmental data. Clarity over the functioning of dengue surveillance might help to improve data reporting, optimize resources and ultimately prevent the spread of outbreaks. The underlying research questions are: how is the dengue surveillance system in District Federal functioning? Which are the main strengths and weaknesses of the dengue surveillance system?

## 2. Materials and Methods

Qualitative data deriving from interviews with 17 experts (see Table 1) have been collected. These were supplemented by locally published guidelines and bulletins. To identify the role of the interviewees, ID codes are used and listed in Table 1. 

Semi-structured interviews [39,40] were used for data collection, and theoretical and purposive sampling were used to select interviewees [41,42,43]. The sample size was determined by data saturation. All the interviews were conducted in Portuguese, then transcribed and translated to English by the authors. Table A1 provides an outline of the interviews, whereas Table A2 lists all the translated Portuguese names.

Thematic analysis was used for data analysis. The main themes emerged were listed in the result section, and further explained in relation to each surveillance activity. To do that, the descriptions of the surveillance activities obtained from the interviews and the main challenges connected to them were grouped and presented in the result section, where they were organized and underpinned using the McNabb’s framework for evaluating surveillance systems [44]. The main themes were then discussed in the discussion and conclusions.

Qualitative information is complemented by quantitative data from the SINAN dating from 2015 to 2019, which were utilized to describe the main sources of notification and the content of the surveillance data. The use of quantitative data is merely for descriptive statistics, and the article does not provide any inferential statistical analysis. Qualitative research with expert interviews and its integration with quantitative data have proven to be useful in the evaluation of health service functioning, including surveillance activities [45,46,47].

Ethical clearance was granted from the ethical committee of Fiocruz Brasilia under registration number 16460819.9.0000.8027 and from the ethical committee of Maastricht University under registration number FHML/GH_2019.076. The interviews were conducted upon receiving the informed consent of the participants and were anonymized for analysis. For the quantitative analysis, anonymous public data from the SINAN were used [29].

## 3. Results

The main themes that emerged are: efficiency of the online reporting system, expertise of the professionals related to surveillance, a lack of resources, underreporting of the private sector, integration between the different domains of surveillance. These themes will be discussed in relation to the functioning of each surveillance activity, which will be explained in detail after a general overview of the surveillance system.

### 3.1. General Overview of Dengue Surveillance in the DF

To provide an overview, a system map based on the McNabb framework [44] was created (Figure 1). 

Suspected cases of dengue are identified and notified from the healthcare units (Figure 1a) through the online portal or paper forms (Figure 1b). Laboratory confirmation is conducted by the Central Laboratory of Public Health (Figure 1c), and the data are analyzed by the Directory of Epidemiological Surveillance (Figure 1d). The Directory of Epidemiological Surveillance is also responsible for response activities (Figure 1e), along with the Directory of Environmental Surveillance, which conducts vector control (Figure 1e,f). The model also shows support activities, which are conducted by the Ministry of Health (Figure 1g).

The DF is subdivided into seven health regions, which are under the responsibility of the seven Regional Superintendencies of Health. Each superintendency has a Unit of Epidemiological Surveillance and a central laboratory of the health region (Figure 1c,d).

### 3.2. Case Detection

As dengue has been endemic in Brazil for two decades, healthcare professionals are experienced in identifying dengue cases. The co-circulation of other arboviroses, with the necessity of a differential diagnosis, is a potential interference for case detection.

**A4**: Good sensitivity, completeness, is what we perceive in practice. Since 2014, the circulation of other viruses in our country (Zika and Chikungunja) has hindered this sensitivity […]. But in terms of capturing information on what we thought dengue was, by 2014, sensitivity was good.

The main obstacle for case detection is the patient’s health seeking behavior. The perceived low quality of health services, the long waiting lines, especially during the epidemics, and the knowledge of the basic treatment for dengue discourage patients from seeking medical attention. Many people prefer to treat the disease at home, leaving many cases unreported because they were unattended by the healthcare system.

**C**: we can’t get those cases who cannot get medical attention in one moment and they end up not seeking again… Because they know all the recommendations for dengue, resting, hydration, they prefer to go back and self-treat at home.

### 3.3. Registration and Reporting of the Cases

Healthcare units must notify every suspected case. The notification form can be filled online on the SINAN portal or, if the unit does not possess a computer, on paper, and consists of 71 items to fill. Once the notification is in the portal, it is accessible by the surveillance units of the municipality, state, and federal government. The online system is a major strength for surveillance.

**A4**: ...it is an easy-to-operate system, has a certain stability, so you can download databases, do analysis easily. As it’s online, allows the managers of the 3 federative entities to make quick decisions.

The suspected deaths for dengue are of immediate compulsory notification to all the management spheres of the health system within 24 hours, through the fastest available means of communication.

Obstacles for notification are the perception that notifying is difficult and time-consuming, the necessity of papery notification due to the lack of technology, the presence of too many items to fill, unawareness of the importance of notification, and the lack of training.

**A1**: (a part is) underreported because the professional doesn’t really notify: it makes the diagnosis, but he does not incorporate the habit of taking a form and informing the surveillance system […] The professional finds it difficult to notify as well [...]. Another issue is that the health service really doesn’t understand the importance of notifying.

Interviewees reported that many forms are incompletely filled, mainly due to the high number of items and to the delays in case confirmation from the laboratories (see section C). This is in line with the numbers of the SINAN database, where the result of the viral isolation was notified in 49% of tested cases, disease evolution was notified in 67% of cases, and diagnostic criterium was specified in 85% of cases.

**A5**: ...and has a lower than expected number of completeness of these important variables, […] because it is so complex, because the fields in the notification form are very large and it ends up being complex to fill. This individual is notified and then it takes time for the investigation…

Through the analysis of the SINAN database, various types of notification units were identified. The percentage of dengue cases notified by each type of unit from 2014 to 2019 in the DF is shown in Table 2. The notification system differs depending on the type of notification unit, and will be described in the following paragraphs. The main types of notification units are: public hospitals and emergency care units (Section 3.3.1), basic healthcare units (Section 3.3.2), private hospitals and laboratories (Section 3.3.3).

#### 3.3.1. Public Hospitals and Emergency Care Units

In the DF, there are 17 public hospitals and six emergency care units, which reported, respectively, 56.32% and 17.65% of cases, making it them the main sources of notification (Table 2). Their notification system is efficient, with the presence of specifically trained health workers assigned to this task. Each hospital has a Unit for Epidemiological Surveillance functioning from 7:00 to 19:00, Monday to Friday, which checks the hospital records for the cases of compulsory notifiable diseases.

**A1**: Inside hospitals, there are Epidemiological Surveillance Units that work with an active surveillance type. [...] So, as a nurse in these centers, I will go through the pediatric emergency, the clinical emergency… […] and I go looking for cases that were suspected for dengue…

#### 3.3.2. Basic Healthcare Units (Unidades Básicas de Saúde (UBS))

With the Family Health Strategy, created in 2006 and implemented in the DF in February 2017, the UBS are the first point of access to the public health system and are responsible for surveillance. Every non-severe case should be treated in the UBS, which must notify it. Therefore, most dengue cases should ideally be reported from the UBS.

There are 172 UBS in the DF, which reported 14.72% of the notified cases (Table 2). According to the interviewees, many UBS were not prepared for the task of surveillance when the Family Health Strategy was implemented. The main problem reported is the lack of technological and human resources. Normally, UBS do not possess a computer or a stable internet connection, and not all the UBS have a medical doctor or enough nurses.

The absence of computers delays the notification process. The report is done on paper forms, which are brought to the regional units of epidemiological surveillance to be digitized. This delays data collection, impeding a timely response to epidemics, especially during the epidemic period, when the UBS are overloaded. The perceived low quality of the UBS and the waiting lists also deter patients from seeking healthcare in the UBS.

**C**: ...and when the case is more serious and he goes to the hospital, theoretically he has already been treated and notified in a UBS, but in reality in these places, especially in the epidemic period, there is a lot of work, so often the professionals don’t have time to fill in all those forms with so many items and the forms accumulate there... often when they go to notify […] it’s been a week or longer.

**B1**: In addition to technological issues, that not all units have a computer [...] we have issues related to the amount of staff that is insufficient. Depending on the volume of cases that arrive at the unit, they can’t do everything, they have to choose: either make the notification, or attend to the patient.

#### 3.3.3. Private Healthcare Units

Private healthcare facilities in the DF notified 9.16% of dengue cases (Table 2). Public hospitals notify, on average, 11.5 times more cases than their private counterparts.

Despite the fact that the notification of suspected cases is compulsory for both public and private facilities, there is a general agreement that the private sector is underreporting. The reasons listed are a lack of a collective health mentality, a lack of integration between public and private, a profit-driven mentality, and cost, as notification requires training and recruitment of dedicated professionals.

**A1**: Public and private should come closer. It should be understood as a single health. The private works a lot with a profit logic. So, for the private sector to put a nurse to do only… not only, but in addition […] she will also notify… if it’s costly for the private sector…. it is of no importance to them, because they don’t see the results either.

Interviewees stated that private hospitals often rely on private laboratories for case notification. Public laboratories cannot notify dengue cases, as their role is limited to the confirmation of suspected cases (Section 3.3). Instead, in the private sector, the healthcare units often do not notify suspected cases, leaving the laboratories the task of notification, causing delays. This seems to be confirmed by the SINAN data, where private laboratories notified almost the same number of cases as private hospitals (Table 2).

The underreporting of the private sector is a problem for surveillance sensitivity, as patients in the private sector have a distinct socioeconomic profile, which the surveillance fails to detect. This results in an underestimation of dengue incidence in some neighborhoods.

**B1**: This is a major concern for us, because the population that seeks the private sector […] has a different socioeconomic profile, and this may compromise the implementation of responses, the adoption of measures... it is in certain places that the government cannot see the problem…

One interviewee reported a positive cooperation with private healthcare facilities in Belo Horizonte (another Brazilian state), where an agreement with the main private provider improved surveillance sensitivity. In the DF, private sector integration is still insufficient.

### 3.4. Laboratory Testing and Case Confirmation

The guidelines for laboratory testing are differentiated between epidemic and non-epidemic period. During the non-epidemic period, every suspected dengue case must be tested. During the epidemic period, it is recommended to test 100% of the severe cases and 10% of all other cases. Most cases should be diagnosed through the clinical-epidemiological criterium, which is defined as the presentation of dengue symptoms during an epidemic. The collection of blood samples for testing is bound to the conformity of the case-definition criteria, which is the presentation of fever plus two or more symptoms [48]. Precise case-definition and guidelines are considered a strength for the system.

**F**: We have very sensitive case definitions, because it favors the entry of many patients in the system

Basic tests, such as NS1 and serology, are performed in hospitals and emergency units. More advanced tests are performed in the Central Laboratory of the State. There are 27 central laboratories in the country, one for each state plus one in the DF. In these laboratories, every sample collected during the earlier stage of the disease is tested for the serotype, as it can be detected only during the viremic phase (day 1 to 3). Only 10% of the collected samples is eligible for that test, meaning that most of the patients access the health system after the opportune period.

The Central Laboratory is responsible for performing the advanced tests and detect circulating serotypes in the district. Each of the seven health regions of the district also has a regional laboratory, where the UBS can send blood samples for performing basic tests.

Interviewees stated that the guidelines for the epidemic period are not implemented, with blood samples collected in almost 70% of the total cases. This corresponds to the SINAN database, where 69% of the cases are diagnosed through testing. According to the interviewees, this overburdens the laboratories, consumes resources, prolongs waiting lists, and hinders laboratory surveillance of other diseases, in a context with limited resources. The causes for guidelines not being implemented are the expectations of the patients (who often demand to be tested), the nescience of the guidelines, and the co-circulation of other arboviruses.

**D**: The surveillance model needs to be adjusted in the epidemic and inter-epidemic phase. From a laboratory point of view, we are really compromised when we do not define it and use the same model in these two different scenarios. So, you end up overloading the laboratory at certain times [...] we have limited human resources, [...] and we often overburden the surveillance service as much as the laboratory when you do not have this very clear definition regarding the models.

### 3.5. Data Analysis and Feedback

The Directory of Epidemiological Surveillance and each Unit of Epidemiological Surveillance of the health regions have one professional for data analysis for arbovirosis surveillance. The regional units are also responsible for digitalizing the papery notifications. As feedback, the Directory of Epidemiological Surveillance publishes a weekly bulletin during the epidemic period and a fortnightly bulletin during the inter-epidemic period. The competence of the professionals working with surveillance is perceived as a strength.

**B1**: Although we have minimal human costs, what we see is the knowledge of each professional.

Digitization of papery forms is a major obstacle for data analysis as it greatly increases the workload of the units, which lack human resources. Interviewees stated that often the notification forms accumulate, and the time spent for digitization is subtracted from data analysis.

**F**: Piles of forms accumulate and sometimes there is one person to type hundreds... and if the notification is delayed, the vector control action later, and everything else, is also delayed.

### 3.6. Role of the Federal Government and Support Activities

The federal government coordinates all the surveillance activities through the Secretary of Health Surveillance, under which the General Coordination of Arbovirosis Surveillance falls. Other departments coordinate other aspects of dengue surveillance, such as: the General Coordination of Public Laboratories, the General Coordination of Public Health Emergencies, and the General Coordination of Environmental Surveillance. The Secretary of Health Surveillance, through its departments, elaborates guidelines, analyses data, produces publications [48,49,50] and, when required, supports the states during epidemics. An epidemiological bulletin on arboviroses is published weekly during the epidemic period and fortnightly during the non-epidemic period.

The General Coordination of Public Health Emergencies collaborates with the arbovirosis surveillance in three ways: elaborating the contingency plan; providing an emergency response unit to implant emergency operation centers; providing emergency trainings for health workers.

### 3.7. Public Health Action

Guidelines for response activities are listed in the contingency plan [49]. Response activities in the DF are coordinated by the Directory of Epidemiological Surveillance through the “District Committee of Coordination and Control” (SDCC) [51]. The SDCC is a committee formed by representatives from various administrative bodies, with the purpose of developing an intersectoral approach for arboviruses control. It consists of weekly meetings with representatives of the fire department, urban cleaning service, companies connected to water and sanitation and infrastructure, and secretaries of education, infrastructure, urban development, and health logistics. The Geiplandengue (regional management for dengue control) has the same structure as the SDCC reproduced in the health regions and implements the decisions of the SDCC. This system, which started in 2016, is fundamental for a comprehensive and integrated response.

**B2**: Inside the SDCC, is also important to highlight all the communication work that is done. Because, as there are several undersecretaries, each secretariat has its communication core […]. Within the epidemiological surveillance only, we would not have this strength. And what is important in this context is the integration…

A problem for the execution of response activities is the turnover of professionals. The management body of surveillance changes according to the local political elections, hindering the development of a consistent strategy. Suggested improvements are the participation of members of the district government to the SDCC, in order to facilitate funding allocation and decision-making, and access to the management positions through public competition.

**E**: there is a high discontinuity of professionals in the field. So, you capacitate, but you have no guarantee that these professional will remain […]. When there is a management change, also a big part of the agents changes.

Politics also interferes when politicians prioritize measures that have greater visibility among the population, rather than carefully planned actions.

**A6**: And dengue in Brazil is very targeted, because it has much... it affects politically [...] When there was an epidemic, the population complained, complained, and the major ordered the ULV, a machine that goes in the car and fogs Ultra Low Volume. But [...] for dengue control program, the ULV is recommended only in extreme cases, and in many places we saw that ULV use was triggered by policy. To say that they were doing something.

### 3.8. Vector Surveillance

The Directory of Environmental Surveillance is subdivided in 15 regional units. One of the services provided are the home visits, aimed to prevent the spread of arboviruses and other vector-borne diseases. This is done by checking the presence of mosquito breeding sites and other vectors. The detected breeding sites and the potential still water containers are eliminated or treated with insecticide. Home visits also have the educational purpose to teach prevention.

The main problems described are: an insufficient number of field workers; difficulties in raising awareness among the population; presence of dangerous neighborhoods.

**H2**: First, I think we would need an exact number of people, a right number to cover the entire region, and to work better on environmental education.

**H1**: Residents always accumulate a lot of water from day to day, 80% of the breeding sites in the DF are inside the homes.

During the home visits, mosquito larvae are collected to produce the Quick Index Survey for Aedes Aegypti. This consists of a quarterly larval sampling of A. Aegypti to estimate the infestation level. The index is used for directing vector control actions, as it presents the magnitude and distribution of mosquito infestation by type of breeding site.

The Environmental Surveillance is also responsible for social mobilization activities, which are planned by a dedicated unit that organizes events in theaters, schools, supermarkets, and other places to teach prevention. This aspect is perceived as crucial and many interviewees argue that it should be more extensive and systematic, involving influential people and institutions, such as bloggers, community leaders, and churches.

**H2**: I think if we worked in community, schools, I think we would cover a larger number, because we would even need fewer people to work with.

The Directory of Environmental Surveillance also participates in response activities, which may be triggered by the epidemiological surveillance or by the environmental surveillance itself

An often-highlighted problem is the lack of integration between epidemiological and environmental surveillance. Reported areas of high dengue incidence often do not correspond to the areas with high infestation indexes. This is due to a lack of joint data analysis, a lack of communication between institutions, the physical separation of the offices, and a non-integrated database.

**A2**: We have different laboratory and entomological systems, and they don’t talk. So, we have to take, almost manually, the base of these two systems and integrate. This, I think, is a very bad factor.

### 3.9. Active Surveillance

Active surveillance exists in the forms of active case finding (hospital unit for surveillance, Section 3.3.2) and syndromic surveillance. Syndromic surveillance is conducted on a federal level in the forms of media monitoring, through the collection of mass-media information about dengue-related events. Another project under development, the Observatorio Dengue, consists of social media monitoring through specific algorithms that correlate data from social media with epidemiological data.

## 4. Discussion

From the interviews, it is possible to infer that the Brazilian surveillance system for dengue is generally perceived as well articulated, sensitive, and robust. The main strengths of the system are considered to be: the experience of healthcare and public health professionals; the simplicity of the online notification system; a direct involvement of the population through home visits and social mobilization activities; the decentralization of surveillance; a sensitive case definition and precise guidelines; and extensive environmental surveillance.

Our analysis revealed that several improvements have been carried out in the last decade to improve the surveillance system. The main improvements mentioned are: the digitalization of the notification system, which significantly sped up surveillance; the introduction of the SDCC, which promoted an integrated response to the epidemics; and the progressive decentralization of the surveillance and response system, which is currently carried out by the municipality.

A common element that emerged is the delay occurring in different phases of surveillance. Timeliness is an important aspect of surveillance and delays in surveillance impair the implementation of an effective response to epidemics. According to the collected data, response activities often occur when the epidemic has already begun. The delay can be attributed to two components: a lack of resources in the surveillance units and impairments in the notification process.

Scientific literature on the topic suggests that investment on surveillance in Brazil is insufficient [22,23], which can be confirmed by this study. From the collected information, an overload of work in the surveillance units seems to occur during the epidemic period, and a lack of human and technological resources makes the implementation of a timely response impossible. Stronger investments in health surveillance might expedite response implementation and help in preventing dengue epidemics before their spread.

Delays in the notification process occur during every phase of data collection. The first delay is caused by the health seeking behavior, as approximately 90% of patients reach the healthcare system during the convalescence phase, according to the interviewees. The second delay occurs during access to healthcare, caused by long waiting lines; this, among other problems, also deters the population from seeking medical attention, leaving part of the cases unreported. The third delay occurs in the notification, due to the overload of healthcare units as well as the necessity—in the UBS—to notify the case on paper and subsequently send it to the epidemiological surveillance to be digitized. The fourth delay, during the confirmation phase, is caused by the overburden of laboratories, which is considered by some interviewees to be the main weakness of the surveillance system. The last delay is in the completion of the forms. Due to the long wait of laboratory confirmation, many professionals forget to complete the notification form when they receive the test results.

To overcome these issues, several improvements have been suggested. An ulterior simplification of the notification sheet could accelerate the process of notification and reduce the number of incomplete reports. Frequent suggestions concerned the creation of technical tools to facilitate the notification process, such as a mobile application, or the integration of the SINAN with the digital records of healthcare units and with the database of public laboratories. To prevent the overload of laboratories, the implementation of the guidelines for the epidemic and non-epidemic period was suggested.

In the case of notification delays, the main challenges for the surveillance system appear to be connected to the lack of resources and the need to improve access to public healthcare. The lack of human resources and up-to-date technology in the healthcare units, the necessity for the healthcare system to cut down waiting lines and reach a bigger part of the population, and the need to improve laboratory efficiency, are the expression of a general necessity of larger investments in public healthcare. To further improve the surveillance effectiveness, it is important for the healthcare system to reach the entirety of the population, particularly in a disease such as dengue, whose socioeconomic component is widely recognized [36,37].

Another highlighted issue regarded the role of the private healthcare system. Official data from the Ministry of Health [52] declared that 30.85% of the DF residents have private health insurance. The comparatively low number of cases reported by private hospitals and the reports of the interviewees seem to indicate private sector underreporting. However, the comparatively low number of cases reported might also be due to other factors, for example, a lower incidence of dengue among patients who rely on private healthcare, rather than underreporting of the private units. As a qualitative study, this work cannot provide statistical evidence of that.

The alleged private underreporting hinders the effective and timely implementation of response activities in specific regions of the district. In an increasingly interconnected society, with daily dislocation of the population in different parts of the city, this does not only affect the areas of high socioeconomic status, but the whole district population.

Better integration of surveillance activities appeared to be another challenge. Interviewees stated that there is an evident need for better integration between environmental and epidemiological surveillance, between surveillance and healthcare, and between different domains of the public administration. An involvement of the general population in dengue prevention is also considered essential. Positive experiences, such as the SDCC, indicate that an integrated response to epidemics is more effective than tackling them merely through the healthcare system, highlighting the need for a comprehensive approach to public health issues.

These results are supported by recent scientific literature, which is increasingly recognizing the necessity of a multisectoral approach for the prevention and control of emerging infectious diseases [53,54]. The WHO is also highlighting the importance of an integrated strategy to improve the outcomes of healthcare systems in infectious disease control [6,55]. This is especially relevant in a disease such as dengue, whose dissemination patterns are influenced by numerous, multisectoral aspects [21]. Systemic approaches are also important for Health Geography, as the understanding of diseases’ spatial distribution requires synergy between different domains of surveillance, environmental and public health [8].

The study has several limitations, and results should be interpreted with caution. The lack of neutrality and objectiveness is intrinsic to qualitative studies [56] and might hinder the conclusions of the research. The selection of experts was also based upon recommendation, possibly introducing selection biases. The study was set in the DF, which is the only Brazilian state with no municipality; the functioning of the surveillance system, with different roles undertaken by municipality, state and federal government, might differ in other Brazilian states. Finally, as a qualitative work, the study does not entail any inferential statistical analysis, and does not provide any quantitative evidence of the described results. The use of numeric data from the SINAN is only intended to describe the sources and the content of the data collected by the surveillance system.

## 5. Conclusions

Brazil’s long history with dengue control and the high incidence of arboviroses led to the development of an articulated and multi-faceted surveillance system, which tackles many aspects of the disease control and prevention. According to our analysis, the main problems for the surveillance system occur when, due to lack of resources, the public healthcare system fails to meet the needs of the population, rather than being caused by a failure of the surveillance system itself. The need for greater public health investment is corroborated by the existing literature, which suggests that public underfunding hinders the access to public healthcare and its effectiveness [32,33]. Investments in public healthcare and surveillance and the provision of enough human and technological resources might accelerate the surveillance processes. This would help to direct timely responses, preventing the spread of disease and saving the resources spent to tackle epidemics.

Another recurring concept of the study is the one of integration, which is considered fundamental for effective surveillance. Integration should be implemented between the different aspects of surveillance, as well as between the different actors of the health system and public administration, in line with a comprehensive concept of healthcare.

In conclusion, to further improve the surveillance system, a more inclusive vision of health should be promoted. This could take into consideration prevention, universal healthcare, and all the intersectoral aspects of dengue. 

## Figures and Tables

**Figure 1 ijerph-17-02062-f001:**
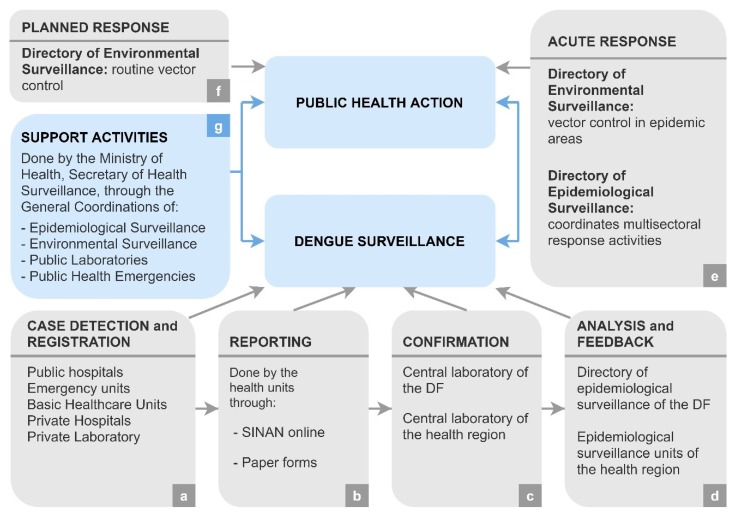
System map of dengue surveillance in the Federal District (Brazil). Suspected cases of dengue are identified and notified from the healthcare units (**a**) through the online portal or paper forms (**b**). Laboratory confirmation is conducted by the Central Laboratory of Public Health (**c**), and the data are analyzed by the Directory of Epidemiological Surveillance (**d**). The Directory of Epidemiological Surveillance is also responsible for response activities (**e**), along with the Directory of Environmental Surveillance, which conducts vector control (**e**,**f**). The model also shows support activities, which are conducted by the Ministry of Health (**g**).

**Table 1 ijerph-17-02062-t001:** List of participants in the study. The identification (ID) code has been used to refer to the content of the interviews in the result section.

Department	N. of Interviewees	ID Code
General Coordination of Arbovirosis Surveillance, Ministry of Health	6	A1–A6
Directory of Epidemiological Surveillance, Federal District	3	B1–B2
Unit of Epidemiological Surveillance, health region of the DF	1	C
General Coordination of Public Laboratories, Ministry of Health	1	D
General Coordination of Public Health Emergencies, Ministry of Health	1	E
General Coordination of Environmental Surveillance, Ministry of Health	1	F
Directory of Environmental Surveillance, Federal District	2	G1–G2
Field workers for environmental (vector) surveillance, Federal District	2	H1–H2

**Table 2 ijerph-17-02062-t002:** Notified dengue cases between 2015 and 2019, per type of notification unit. Data source: Sistema de Informação de Agravos de Notificação (SINAN).

Type of Notification Unit	% of Notified Cases
Public Hospitals	56.32
Emergency Healthcare Units	17.65
Basic Healthcare Units	14.72
Private Hospitals	4.32
Private Laboratories	4.08
Non-Governmental Organizations	1.34
Other Public Units	0.80
Other Private Units	0.73

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
