# Peer review of "Dengue Surveillance System in Brazil: A Qualitative Study in the Federal District"

_ijerph, 2020, doi:10.3390/ijerph17062062_

Round 1

Reviewer 1 Report

Thanks for the opportunity in reviewing this work.

Summary

The paper “Dengue surveillance system in Brazil: a qualitative study in the Federal District " addresses the problem of exploring the strengths and weaknesses of the Brazilian surveillance system, particularly looking at the functioning of data collection and reporting.

Overall, I think the paper is well written and is a nice contribution to the public discussion. Nevertheless, I have the feeling that some points need be carved out.

MAIN POINTS

- Section 1.2 line 110 – This is not a political journal. I would advise authors to refrain phrases like this one: “Private sector underreporting is particularly relevant in a country where austerity measures are resulting in an increasingly privatized healthcare system ⁠with a growing number of private-public partnerships (…) even in poor urban settings.”

For example, the sentence in the discussion section “Scientific literature on the topic suggests that investment on surveillance in Brazil is insufficient (22,23), which can be confirmed by this study. (…) Stronger investments in health surveillance (…)” is a good example of how the authors should write their statements.

- section “Discussion”: the lines 532 and 537 seem to be in contradiction. This contradiction is noted by the authors themselves in line 561 “Finally, private sector underreporting has been deduced from the interviews, and the study does not provide statistical evidence of this phenomenon.”

- The paper lacks a real statistical analysis.

Author Response

Dear Reviewer,

Thank you for the valuable suggestions you provided. Several modifications have been made to the manuscript in order to improve the flow and the legibility of the paper, especially in relation to the result section.

  1. Section 1.2 line 110 – This is not a political journal. I would advise authors to refrain phrases like this one: “Private sector underreporting is particularly relevant in a country where austerity measures are resulting in an increasingly privatized healthcare system ⁠with a growing number of private-public partnerships (…) even in poor urban settings.

The sentence has been rephrased in way that entails less political implications. The sentence was written with the intention of highlighting the relevance of the paper, and does not entail any political position from the Authors.

  1. section “Discussion”: the lines 532 and 537 seem to be in contradiction. This contradiction is noted by the authors themselves in line 561 “Finally, private sector underreporting has been deduced from the interviews, and the study does not provide statistical evidence of this phenomenon.”

The sentence has been rephrased in a way that clarifies how the paper does not provide any statistical analysis, and the low number of cases reported by the private sector might be caused by other factors.

  1. The paper lacks a real statistical analysis.

As a qualitative paper, the work lacks an inferential statistical analysis. In the revised version, lack of inferential statistical analysis has been specified more clearly in the limitations. Furthermore, the use of numeric data from the SINAN – which is only used to describe the sources and the content of the data collected by the surveillance system – has been described in the methodology in a clearer way.

I believe that the modification performed have improved the conclusions of the study.

Furthermore, you observed that the presentation of the results can be improved. In order to do so, several modifications have been executed.

  • The result section has been reduced by 25% (from 4411 to 3321 worlds), to improve the legibility and the flow of the paper.
  • The result section starts now with an overview of the general themes emerged from the thematic analysis. This is done to show how each of the themes emerged is related to the different surveillance activities.

I hope I answered sufficiently to all your observations regarding the paper. I thank you again for reviewing my work. I am available to provide further clarifications if needed.

Reviewer 2 Report

Reviewer’s Comments:

This study aimed to explore the strengths and weaknesses of the Brazilian dengue surveillance system, particularly looking at the functioning of data collection and reporting through semi-structured interviews with experts in dengue surveillance, supported by quantitative data from the official notification system. In general, the paper is well-written, but this reviewer thinks that results section is too lengthy and need to be summarized. I also have a few more comments.

53: Please correct this sentence: “In Brazil, dengue usually occurs in cities with more than 500,000 inhabitants, highlighting the necessity for an adequate and effective urban surveillance system.” You may say that, in Brazil, dengue incidence is greater in cities with more than 500,000 inhabitants…….

70: Please expand PCR.

105: Is there a geographical variability in underreporting according to this study? “A recent study estimated that, for each 12 cases of dengue, only one was reported to the SINAN.”

151: Please provide the outline of the semi-structured interview (as a supplement).

159: What were the themes (topics, ideas and patterns of meaning that come up repeatedly) emerged in the analysis?

Discussion: Please discuss more about improvements in the dengue surveillance system over past years (or decades).

General Comment: Please reduce the results by summarizing details.

Author Response

Dear Reviewer,

Thank you for the valuable suggestions you provided. Several modifications have been made to the manuscript in order to improve the flow and the legibility of the paper, especially in relation to the result section.

  1. General Comment: Please reduce the results by summarizing details.

The results have been cut down by 25% (from 4411 to 3321 worlds). I believe that the content of the result section should not be further reduced, as one of the declared objectives of the study is to explain in detail the functioning of the surveillance system for dengue. However, I also believe that, as suggested, the reduction of the result section improved the general quality of the paper, making it more easy to read.

  1. 53: Please correct this sentence: “In Brazil, dengue usually occurs in cities with more than 500,000 inhabitants, highlighting the necessity for an adequate and effective urban surveillance system.” You may say that, in Brazil, dengue incidence is greater in cities with more than 500,000 inhabitants…….

The sentence has been corrected accordingly to the suggestion provided.

  1. 70: Please expand PCR.

In the revised version, PCR has been briefly explained, mentioning the detection of virus RNA, the time window in which the test can be performed, and the possibility of serotype detection with the technique.

  1. 105: Is there a geographical variability in underreporting according to this study? “A recent study estimated that, for each 12 cases of dengue, only one was reported to the SINAN.

The study mentioned was set in Salvador, a Brazilian city in the state of Bahia. No geographical variability was reported in the study. The geographical location of the study has been added to the reviewed manuscript.

  1. 151: Please provide the outline of the semi-structured interview (as a supplement).

The outline of the interviews has been added to the supplementary material. This was mentioned in the method section.

  1. 159: What were the themes (topics, ideas and patterns of meaning that come up repeatedly) emerged in the analysis?

The result section starts now with an overview of the general themes emerged from the thematic analysis. I believe this suggestion improved the overall clarity of the methodology and result presentation, as it is now shown more clearly how all the themes emerged during the analysis relate to the different surveillance activities.

  1. Discussion: Please discuss more about improvements in the dengue surveillance system over past years (or decades).

Improvements in the dengue surveillance system mentioned during the interviews have been added to the discussion.

Furthermore, you observed that the research design and the methodology description can be improved. In order to do so, several modifications have been done.

  • The use of quantitative data has been described more clearly, specifying that quantitative data are merely used to describe the data collected by the surveillance system (descriptive statistics) and that the work does not entail any inferential statistic.
  • The method for analysing qualitative data has been explained more in detail.

I hope I answered sufficiently to all your observations regarding the paper. I thank you again for reviewing my work. I am available to provide further clarifications if needed.

Reviewer 3 Report

The authors seek to evaluate the epidemiological surveillance system of dengue in Brazil, using interviews with experts and those responsible for applying the surveillance system in the country. The study design is adequate, and the objectives are very clear.

The authors refer to having used the qualitative method in the collection and analysis of information, adding data of a quantitative nature.

The method of gathering information seems to be correct, although the way of analysing it may need to be improved and clarified, it is not clear how the interviews were treated, in order to obtain the results and conclusions that are presented in the article.

Author Response

Dear Reviewer,

Thank you for the valuable suggestions you provided. Several modifications have been made to the manuscript in order to improve the way in which interviews have been analysed in order to obtain the results and conclusions that are presented in the article.

The method for analysing the collected data has been explained more in detail in the methodology section. Several themes were extrapolated by the analysis of the interviews. These themes were further explained in relation to each surveillance activity.

Moreover, the result section starts now with an overview of the general themes emerged from the thematic analysis. This is done to show how each of the themes emerged is related to the different surveillance activities.

To further improve clarity regarding data collection, the outline of the interview questions was also added to the methodology.

The use of quantitative data has also been described more clearly, specifying that quantitative analysis was merely used to describe the data collected by the surveillance system (descriptive statistics), and that the work does not entail any inferential statistic.

Furthermore, you observed that the results presentation and the methodology can be improved. In order to do so, several modifications have been done.

  • The result section has been reduced by 25% (from 4411 to 3432 worlds), to improve the legibility and the flow of the paper.
  • In the discussion, the observation regarding underreporting of the private sector (line 1517) and the limitations of the study (line 1542) better explain the use of descriptive statistics.

I hope I answered sufficiently to all your observations regarding the paper. I thank you again for reviewing my work. I am available to provide further clarifications if needed.

Round 2

Reviewer 2 Report

I think that authors have addressed my concerns. I have no further comments.

Author Response

Dear Reviewer,

Thank you for reviewing my paper and for your valuable feedbacks.

I have no further comments.

Reviewer 3 Report

The authors made important clarifications in the article, however it remains unclear how the results are extracted from the interviews.

Author Response

Dear Reviewer,

The process of extraction and presentation of the results has been further clarified in the methodology section.

Thank you for reviewing the paper and for your valuable feedbacks.